# Current practice patterns of osteoporosis treatment in cancer patients and effects of therapeutic interventions in a tertiary center

**Nasa Fujihara**[1☺*], **Yuki Fujihara**[2☺], **Shunsuke Hamada**[1‡], **Masahiro Yoshida**[1‡], **Satoshi Tsukushi**[1☺]

**1** Section of Orthopedic Surgery, Aichi Cancer Center, Nagoya, Japan, **2** Section of Orthopedic Surgery, Nagoya Ekisaikai Hospital, Nagoya, Japan

☺ These authors contributed equally to this work.
‡ These authors also contributed equally to this work.
* nfujihara@aichi-cc.jp

**Data Availability Statement:** Data are available from the ACC Institutional Data Access / Ethics Committee (contact ortho_tiken@aichi-cc.jp,

## Abstract

Cancer and osteoporosis have high incidence rates in older populations. However, the treatment of osteoporosis among cancer patients has not been adequately described. Our purpose was to clarify the current practice patterns of osteoporosis treatment among cancer patients in an academic cancer center, and to analyze the efficacy of treatment interventions. Patient records from April 2009 to March 2018 were retrospectively reviewed, and the study included a total of 316 cancer patients with osteoporosis. After patients' data extraction, the patients were divided into two groups, with (n = 144) or without treatment (n = 172), and compared the outcomes of these groups to evaluate the medication effect. The primary outcome was new radiographic fragility fractures during the study period. The related factors associated with fracture injuries and the rate of adverse events, such as osteonecrosis in the jaw and atypical femoral fractures, were analyzed. The rate of treatment intervention was 45.6% among the patient groups. Among patients in the study group, breast cancer patients (n = 107) were mostly treated (n = 79, 73.8%) with oral bisphosphonate. A significant difference in new fracture rate was observed between the two groups (treatment group, 30.6%; non-treatment group, 54.7%), and the risk of fracture was 42% lower in the treatment group (hazard ratio, 0.58; 95% confidence interval, 0.39–0.86; p<0.05). Previous chemotherapy, steroid use, and older age were significantly associated with increased rate of new fragility fractures. The adverse event rate was 3.5% (presented in five cases). Older cancer patients who receive chemotherapy or steroids are strongly recommended undergo bone quality assessment and appropriate osteoporosis treatment to improve their prognosis.

## Introduction

Treatment of osteoporosis is essential for maintaining a good quality of life in older individuals [1, 2]. Despite preventative efforts, bone fragility increases with age. Therefore, even older people in good health are required to make a steady effort to avoid problems caused by bone

irb@aichi-cc.jp) for researchers who meet the criteria for access to confidential data.

**Funding:** The author(s) received no specific funding for this work.

**Competing interests:** The authors have declared that no competing interests exist.

fragility, which can often be challenging [3, 4]. Additionally, cancer is a well-known disease that occurs in older populations. However, the treatment of osteoporosis among cancer patients has not been adequately described, despite the huge risk of developing osteoporosis. The International Society of Geriatric Oncology (ISGO) group reported that classical factors including age, sex, family history of hip fractures, comorbidities, as well as corticosteroid, tobacco, and alcohol consumption are emerging as prevalent risk factors for osteoporosis in cancer patients [5]. Moreover, another study reported that the severity of bone disease and number of lesions help identify patients who are at risk of bone failures, including fragility fractures [6]. However, few studies have examined the incidence of adverse events associated with osteoporosis in cancer patients with such risk factors.

The ISGO group also reported that even the treatment recommendations are merely a summary of current knowledge and need further discussion because of the lack of study data. Osteoporosis is often misunderstood as a disease that is not fatal, as there are no symptoms in many cases. However, a decrease in the activity levels or immobilization caused by osteoporotic fragility fractures could be a major obstacle associated with cancer chemotherapies and is directly related to the worsening of cancer outcomes [7–10]. Therefore, it is important to prevent osteoporotic fractures in cancer patients. However, few studies have examined the effects of treatment interventions for osteoporosis in cancer patients. Although there is a growing interest in particular fields to preserve bone health, such as bone fragility associated with hormone treatment [3, 11], a comprehensive analysis of the treatment intervention rate for osteoporosis among cancer patients has not been conducted.

This study aimed to provide an overview of osteoporosis treatment efforts implemented by our center hospital institution for cancer treatment and to identify the effect of preventative medication on new fragility fractures. We expected to observe a higher intervention rate of osteoporosis treatment in patients who received hormone therapy, such as breast or prostate cancer patients. We also hypothesized that the therapeutic intervention would be effective in preventing new fragility fractures, as observed in previous studies [12, 13], and if a similar therapeutic effect is observed in cancer patients, then treating osteoporosis may also improve cancer treatment outcomes.

## Materials and methods

### Study design and participants

This retrospective study was approved by Aichi Cancer Center institutional ethics committee (approval number 364). After approval, data from our center's medical records from April 2009 to March 2018 were obtained. As this was a retrospective cohort study, the need for informed consent was waived by our ethics committee, but all participants were given the option to decline participation by way of opting out through the website.

Following the Japanese osteoporosis prevention and treatment guidelines of 2015, the inclusion criteria included a history of vertebral or femoral fragility fractures or a dual energy X-ray absorptiometry (DXA)-scanned young adult mean score <70%. This criteria extracted all cancer patients aged >40 years who could be newly diagnosed with osteoporosis in our hospital. The exclusion criteria were history of diagnosis or treatment of osteoporosis before participation, pathological fractures with bone metastasis, age <40 years [14, 15], and follow-up period <6 months.

After extracting data from 316 patients who met the criteria, we investigated the patient demographics, cancer type, cancer treatment, and whether osteoporosis was treated or not. The primary outcome was new radiographic fragility fractures during the study period. To detect fragility fractures, data were reviewed from all imaging studies, including conventional

lateral spine radiographic photos, computer tomography scans, and spinal magnetic resonance imaging, that were conducted during the follow-up period in each patient. For these techniques, evaluation was performed by two independent orthopedic surgeons, and if the evaluation differed between surgeons, senior orthopedic surgeons judged the results. The average frequency of evaluation in each department was 2–3 months. Treatment intervention of osteoporosis was defined as treatment with anti-osteoporotic agents >6-month period, and steroid use was defined when a patient received prednisolone 7.5 mg/day for more than 3 months [16].

## Data analysis

We grouped patients based on whether they received treatment or not. The data of patients treated with oral bisphosphonates (treatment group, n = 101) were compared with the data of those without treatment (non-treatment group, n = 172) to determine the medication effects for preventing new radiographic fragility fractures. The cumulative fracture rate was described using a Kaplan–Meier curve, and a log-rank test was performed to compare the intergroup differences. Additionally, a Cox proportional hazard regression model was applied to evaluate the risk factors associated with new fracture injuries. The factors examined were sex, age, method of diagnosis, cancer type, presence of bone metastases or duplicate cancers, history of chemo/radiotherapy, steroid use, and treatment intervention. Then, a multivariate analysis was conducted with all variables from the univariate analysis. All analyses were performed using STATA/SE version 14.2 (StataCorp, College Station, TX, USA), and the level of significance was set at $p < 0.05$.

## Results

In total, 316 cancer patients (mean age, 70.0 [range 40–93] years; sex, 81 men [25.6%] and 235 women [74.4%]) were diagnosed with osteoporosis during the 10-year study period. The patient demographics are presented in Table 1. Of these, 48 (15.2%) and 29 patients (9.2%) had bone metastasis and double cancers, respectively. Moreover, 188 (59.5%) and 79 patients (25.0%) had previously undergone chemotherapy and radiotherapy, respectively. In relation to chemotherapy, steroid injections were used in 46.8% (n = 148) of patients. The mean follow-up period was 34.2 (range, 6.1 to 114.5) months, and the total number of deaths during the study period was 96 (30.4%).

Patients were mostly diagnosed with breast cancer (n = 107, 34.0%), followed by gastrointestinal (n = 72, 22.8%), lung (n = 47, 14.9%), and blood cancer (n = 42, 13.3%). Contrary to our expectation, prostate cancer patients were almost undiagnosed (<2%). The total rate of medications was 45.6% (n = 144), with 70.1% (n = 101) of patients receiving oral bisphosphonate, followed by 9.7% (n = 14) of receiving zoledronic acid injections. An active vitamin D3 form, the one most commonly prescribed in general practice, was used in 9% of all cases. Denosumab was used in 6.9% of the treatment-group participants. Apart from the healthy population, only one participant (0.7%) received a teriparatide injection at another clinic (Fig 1). Osteoporosis was most frequently treated in breast cancer patients (n = 79, 73.8%), followed by blood (n = 18, 42.9%), lung (n = 14, 29.8%), and gastrointestinal cancer patients (n = 20, 27.8%). However, only 16.0% of head and neck cancer patients received medication for osteoporosis (Fig 2).

The total rate of new fragility fractures during the study period was 43.7% (n = 138) (i.e., 30.6% [n = 44] and 54.7% [n = 94] in the treatment and non-treatment groups, respectively). Although there was clearly a statistically significant difference, only the statistics of the oral bisphosphonate group (n = 101) were examined because of the need for uniformity in the

**Table 1. Patient demographics.**

| Characteristics | Total (n = 316) | Treatment (n = 144) | Non-treatment (n = 172) | *p*-value |
|---|---|---|---|---|
| **Age, years** | | | | |
|     **Mean (range)** | 70.0 (40–93) | 67.4 (50–86) | 71.4 (51–84) | 0.06 |
| **Sex, female** | 74.4% (235) | 88.9% (128) | 62.2% (107) | <0.01 |
| **Diagnosis** | | | | |
|     \*DXA | 32.9% (104) | 56.3% (81) | 13.4% (23) | |
|     **Vertebral fracture** | 60.1% (190) | 38.2% (55) | 78.5% (135) | |
|     **Femoral fracture** | 7.0% (22) | 5.6% (8) | 8.1% (14) | |
| | | | | <0.01 |
| **Cancer** | | | | |
|     **Breast** | 34.0% (107) | 54.9% (79) | 16.3% (28) | |
|     **Gastrointestinal** | 22.8% (72) | 13.9% (20) | 30.2% (52) | |
|     **Lung** | 14.9% (47) | 9.7% (14) | 19.2% (33) | |
|     **Blood** | 13.3% (42) | 12.5% (18) | 14.0% (24) | |
|     **Head and neck** | 7.9% (25) | 2.7% (4) | 12.2% (21) | |
|     **Others** | 7.3% (23) | 6.3% (9) | 8.1% (14) | |
| | | | | <0.01 |
| **Double cancer** | 9.2% (29) | 5.8% (10) | 11.0% (19) | 0.21 |
| **Bone metastasis** | 15.2% (48) | 19.4% (28) | 11.6% (20) | 0.05 |
| **Previous radiotherapy** | 25% (79) | 19.4% (28) | 29.7% (51) | 0.04 |
| **Previous chemotherapy** | 59.5% (188) | 56.2% (81) | 62.2% (107) | 0.28 |
| **Steroid use** | 46.8% (148) | 47.2% (68) | 46.5% (80) | 0.9 |
| **Adverse effect** | | | | <0.01 |
|     **Additional fracture** | 43.7% (138) | 30.6% (44) | 54.7% (94) | |
|     **Jawbone necrosis** | | 2.1% (3) | 0% (0) | |
|     **Atypical femoral fracture** | | 1.4% (2) | 0% (0) | |

\*DXA: dual energy X-ray absorptiometry.

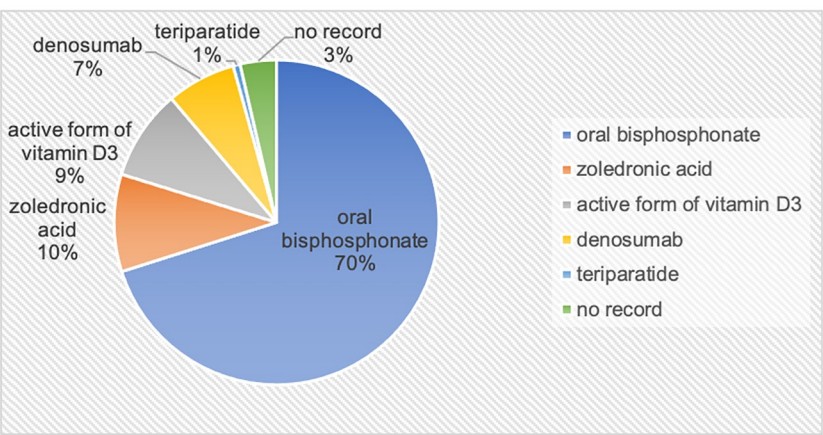

**Fig 1. Key drugs used for osteoporosis treatment and frequency of anti-osteoporotic agents.** Of the oral bisphosphonates, 54.9% were alendronate and the rest were risedronate. The active form of vitamin D3 included eldecalcitol and alfacalcidol.

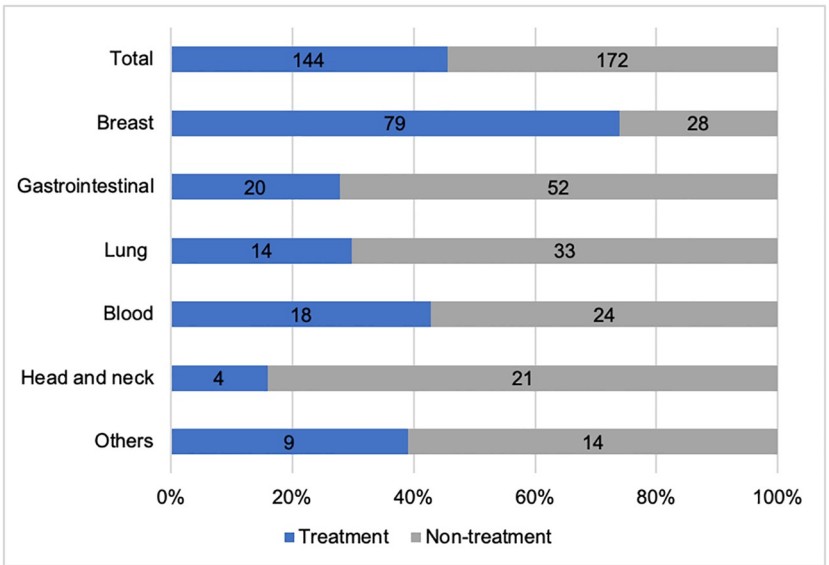

**Fig 2. Treatment intervention rate.** The total rate of treatment intervention was 45.6%, and if breast cancer patients were excluded, it reduced to 31.0%. Approximately 73.8% of breast cancer patients were treated. In contrast, only 16.0% of head and neck cancer patients were treated.

treatment agents. The Nelson-Aalen cumulative hazard estimate showed a great difference in the risk of new radiological fragility fractures between the treatment (n = 101) and non-treatment groups (n = 172), and the log-rank test demonstrated a significant difference among these groups ($p < 0.001$) (Fig 3). However, the greater percentage of DXA-diagnosed patients could be a bias for the treatment group. We also performed the analysis after excluding the DXA-diagnosed patients, but it presented similar results (Fig 4). Although the difference between the two groups was small, the log-rank test also showed a significant difference between the two groups ($p < 0.05$).

The multivariate analysis revealed that previous chemotherapy (hazard ratio [HR], 1.78; $p = 0.019$; 95% confidence interval [CI], 1.10–2.89), steroid use (HR, 1.68; $p = 0.018$; 95% CI, 1.09–2.58), and older age (HR, 1.03; $p = 0.007$; 95% CI, 1.01–1.05) were positively related to the increase in the rate of getting another fracture. Treatment intervention (HR, 0.58; $p = 0.007$; 95% CI, 0.39–0.86) and DXA diagnosis (HR, 0.29; $p < 0.001$; 95% CI, 0.15–0.51) showed an inverse result (Table 2).

The most prevalent new radiological fragility fractures were vertebral (76.8%, n = 106) and femoral fractures (11.6%, n = 16). An adverse effect from using anti-osteoporotic drugs was observed in five cases (3.5%). This included two cases (1.4%) of atypical femoral fractures with long-term use of zoledronic acid, and three cases (2.1%) of jawbone necrosis in relation to the use of zoledronic acid or denosumab injections. Patients with fractures underwent open reduction and internal fixation at our department without any complications. Jawbone necrosis patients were treated at the dental department of our hospital, and all patients recovered during the follow-up period.

## Discussion

Although the prognosis of cancer patients has improved, it is necessary to treat osteoporosis to maintain the performance status (PS) for chemotherapies [5, 17]. With this perspective, we

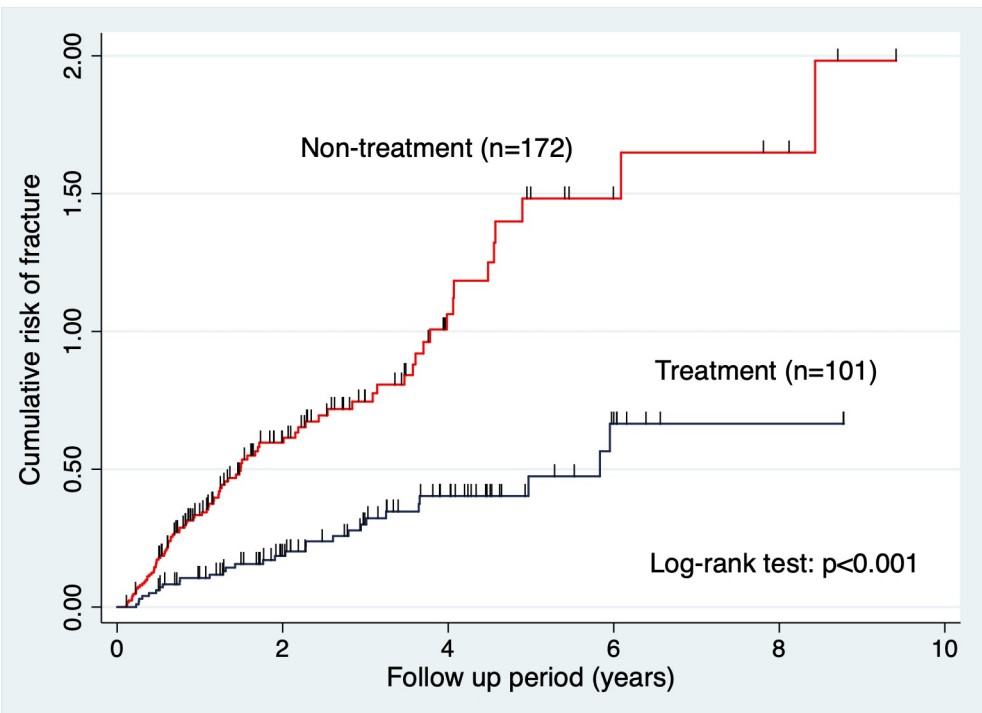

**Fig 3. Cumulative risk of fracture (treatment vs non-treatment group).** The Nelson-Aalen cumulative hazard estimate showing the difference in the risk of additional fragility fractures between the two groups. The log-rank test showed a significant difference between these two groups ($p<0.001$).

found that medication had a significant influence on osteoporosis even in cancer patients. The use of anti-osteoporotic agents could reduce the risk of new radiological fragility fractures by 42%. However, the treatment rate itself was below 50%, despite the existence of the national guidelines. These results were fairly better than those of the general population (i.e., approximately 20% in Japan) [18, 19].

Disparities in treatment interventions between departments were large, with well-conducted interventions in breast and hematology departments, but not in lung, gastrointestinal, or head and neck surgery departments. This may stem from the differences in the cancer treatment itself, although we have not studied the exact reasons for this issue. For example, breast cancer treatment is directly related to bone metabolism with the use of hormone treatment, such as tamoxifen or aromatase inhibitors. There are several guidelines that have already been published for bone-directed treatment, and Hadji et al. [11] have updated the treatment algorithm for better assessment of fracture risk. In addition, Colzani et al. [20] reported an increased risk of fragility fracture in patients using aromatase inhibitors compared with patients using tamoxifen (HR, 1.48; 95% CI, 0.98–2.22). Therefore, in such cases, a treatment intervention should be provided. In these cases, physicians and patients may easily accept osteoporosis treatment because it is already included in the cancer treatment.

Concerning risk factors, chemotherapy and associated steroid use had an obvious effect on the development of new radiological fragility fractures. Although chemotherapy itself is a risk factor for osteoporosis, a certain amount of steroid is often used as an antiemetic in chemotherapy, and this may worsen the patient's comorbidities, which were not considered fatal. Many cancer patients are prescribed more steroids than the diagnosis criteria of steroid-

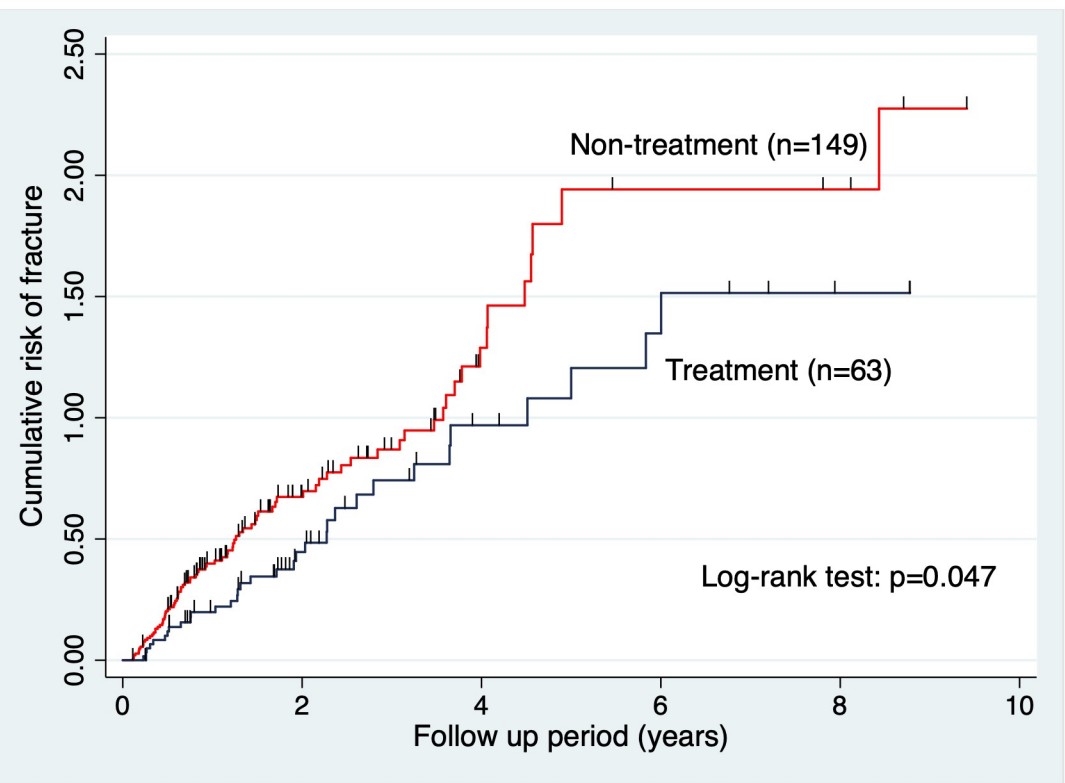

**Fig 4. Cumulative risk of fracture without DXA-diagnosed patients (treatment vs non-treatment group).** When patients diagnosed by DXA were excluded, the difference between the two groups narrowed. However, the log-rank test still showed a significant difference between these groups ($p = 0.047$). DXA, dual energy X-ray absorptiometry.

induced osteoporosis [16], and the risk of developing this condition is dose dependent [21, 22]. In this study, patients who received chemotherapy and used steroids were at an increased risk of developing additional fractures, indicating that more careful interventions are needed for these patient groups.

After considering these results, we propose some suggestions for individual physicians who are engaged in cancer treatments. First, as the use of osteoporotic drugs has an evident effect on preventing new fragility fractures, physicians should take osteoporosis seriously to provide an efficient cancer treatment. Indeed, the treatment of osteoporosis is a time-consuming process, as oral bisphosphonates are sometimes not fully effective in patients who have difficulty with oral intake or are unable to receive oral medication. However, in recent years, several new

**Table 2. Factors related to the risk of new radiological fragility fractures.**

| Factors | Hazard ratio | *p*-value | 95% CI |
|---|---|---|---|
| Previous chemotherapy | 1.78 | 0.019 | 1.10–2.89 |
| Steroid use | 1.68 | 0.018 | 1.09–2.58 |
| Older age | 1.03 | 0.007 | 1.01–1.05 |
| Diagnosed with DXA | 0.29 | <0.001 | 0.15–0.51 |
| Treatment intervention | 0.58 | 0.007 | 0.39–0.86 |

CI: confidence interval; DXA: dual energy X-ray absorptiometry.

anti-osteoporotic agents, such as annual zoledronic acid, semi-annual denosumab, or newly developed romosozumab injections have been released and are gradually being used. These new drugs are potentially therapeutically beneficial and cost effective for cancer patients who must undergo multiple medical treatments [23–25]. The use of appropriate drugs for osteoporosis may also reduce the use of unnecessary analgesics, such as opioids or morphine, which can ultimately lead to a worsened PS. Second, as aforementioned, an increased rate of anti-osteoporotic agent usage can be attained as long as it can be linked to the cancer treatment itself. For example, when the diagnostic criteria for steroid osteoporosis are correctly applied to all cancer patients receiving chemotherapy, osteoporosis treatment could be included in their cancer treatment and physicians may prevent the decrease of the PS by avoiding fragility fractures caused by osteoporosis. Finally, there is an urgent need to correct disparities among medical departments. Common criteria are needed for initiating osteoporosis treatment in an institution. Moreover, establishing a cancer board that includes all departments may also help improve the rate of treatment interventions.

Regarding the adverse effects of using anti-osteoporotic drugs, jawbone necrosis and atypical femoral fracture were observed in 1.4%, and 2.1% of patients, respectively. This was similar to those previously reported by Edwards et al. [26] and Lipton et al. [27]. Our institution routinely requires dental consultations for screening oral conditions before starting anti-osteoporotic agent administration, and patients who were diagnosed with a high risk of jawbone necrosis underwent dental treatment before using these agents [28]. In addition, there were three cases of atypical femoral fractures in our study group, and all were treated surgically at our orthopedic department without any complications.

In this study, we had few patients who used teriparatide, which is rarely used for osteoporosis treatment in recent years. After Subbiah et al. [29] reported the risk of teriparatide-induced osteosarcoma, the use of teriparatide is avoided for cancer patients. However, recent research has provided new evidence concerning the use of teriparatide [25, 30, 31]. Interestingly, Gilsenan et al. reported that the incidence of osteosarcoma associated with teriparatide use during a 15-year period was not different than what was expected based on the background incidence rate of osteosarcoma after examining data from a US database [30]. Besides, other new agents, such as romosozumab, humanized IgG2 monoclonal antibody, or annual zoledronic acid, were allowed to be used for the treatment of cancer patients. These relatively new drugs will be used more aggressively for such patients in the future.

However, this study had several limitations. First, there were differences between the two groups. Although we conducted the analysis without factors that could affect the results, the potential difference may have affected the study results. Additionally, we had no detailed data on cancers and chemo/radiotherapies, and these unknown factors may also have affected the results. Second, we applied Japanese national guidelines for the prevention and diagnosis of osteoporosis, which are slightly different from the World Health Organization guidelines. This may have affected treatment interventions in particular for patients from different departments, such as the breast oncology department. Third, we defined treatment interventions as those provided for more than 6 months, as cancer patients often have poor prognoses. However, most related studies set this intervention period as over 1 year; therefore, some may doubt the effect of treatment interventions.

Despite these limitations, the current study provided a sound analysis of the effect of osteoporosis treatment among cancer patients in our tertiary center. We found that the medication effect was quite substantial, while the incidence of adverse events was low. Therefore, older cancer patients who receive chemotherapy or use steroids are strongly recommended to use anti-osteoporotic drugs to maintain their PS. However, there is still a need to study the effect of osteoporotic treatment in terms of specific cancer types or agents, as related research is

lacking. From these viewpoints, further research is required to improve the quality-of-life associated with cancer treatment of patients to achieve better prognoses.

## Supporting information

**S1 Data.**
(XLSX)

**S1 Checklist.**
(DOCX)

## Acknowledgments

The authors thank Hidemi Ito and Kenichi Yoshimura for their statistical support.

## Author Contributions

**Conceptualization:** Nasa Fujihara, Yuki Fujihara, Satoshi Tsukushi.

**Data curation:** Nasa Fujihara, Yuki Fujihara.

**Formal analysis:** Nasa Fujihara, Yuki Fujihara.

**Investigation:** Nasa Fujihara.

**Methodology:** Nasa Fujihara.

**Supervision:** Shunsuke Hamada, Masahiro Yoshida, Satoshi Tsukushi.

**Writing – original draft:** Nasa Fujihara.

**Writing – review & editing:** Nasa Fujihara, Yuki Fujihara, Shunsuke Hamada, Masahiro Yoshida, Satoshi Tsukushi.

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
