## [Editor Report · Decision Letter 0]

2 Nov 2020

PONE-D-20-33343

Current practice patterns of osteoporosis treatment in cancer patients and effects of therapeutic interventions in a tertiary center

PLOS ONE

Dear Dr. Fujihara,

Thank you for submitting your manuscript to PLOS ONE. After careful consideration, we feel that it has merit but does not fully meet PLOS ONE’s publication criteria as it currently stands. Therefore, we invite you to submit a revised version of the manuscript that addresses the points raised during the review process.

We look forward to receiving your revised manuscript.

Kind regards,

Robert Daniel Blank, MD, PhD

Academic Editor

PLOS ONE

Additional Editor Comments:

There are several important issues that you must address before I send this MS to other reviewers.

1. Please check your MS against STROBE standards for cohort studies and add the STROBE checklist to your MS.

2. Please show p-values for table 1. My inspection of the table suggests that the treated/untreated groups differ. This should be checked explicitly, as it is an important potential source of confounding in your subsequent analysis.

3. Date of entry into study should be explicitly defined, and survival curves plotted based on study entry, not diagnosis of osteoporosis. It is necessary to unpack those who were already being treated prior to cancer diagnosis.

4. What about osteoporosis based on other low trauma fractures beside vertebral or femoral?
---

## [Author Response · Author response to Decision Letter 0]

8 Dec 2020

November, 30, 2020

Dear Editor:

Thank you for checking the manuscript. We have summarized the corrections below based on your review. Thank you in advance for your kind help.

Best regards,

Nasa Fujihara

Additional Editor Comments:

1. Please check your MS against STROBE standards for cohort studies and add the STROBE checklist to your MS.

⇒ The checklist is attached at the end of the manuscript.

2. Please show p-values for table 1. My inspection of the table suggests that the treated/untreated groups differ. This should be checked explicitly, as it is an important potential source of confounding in your subsequent analysis.

⇒ We have included it because it affects the results of the analysis as you pointed out. Although there were fewer women in the untreated group, which may lead to a lower risk of fracture, the results actually showed a higher risk of fracture.

3. Date of entry into study should be explicitly defined, and survival curves plotted based on study entry, not diagnosis of osteoporosis. It is necessary to unpack those who were already being treated prior to cancer diagnosis.

⇒As you pointed out, the starting point was the time of participation in the study, and we have corrected this to include the notation in the graph.

Also, patients with a diagnosis of osteoporosis prior to the start of the study were excluded and this has been noted.

4. What about osteoporosis based on other low trauma fractures beside vertebral or femoral?

⇒ To clarify the definition, we only included the cases with vertebral or low-energy femoral fractures, or less than 70% of YAM value according to the Japanese diagnostic criteria of osteoporosis. So as you pointed out, the other low trauma fracture with high YAM value cases are not included. However, results includes all fragility fractures caused from low energy trauma.

Journal Requirements:

⇒Done

⇒ I have changed the manuscript following above.

b) If there are no restrictions, please upload the minimal anonymized data set necessary to replicate your study findings as either Supporting Information files or to a stable, public repository and provide us with the relevant URLs, DOIs, or accession numbers. 

⇒Done

---

## [Decision Letter · Decision Letter 1]

23 Dec 2020

PONE-D-20-33343R1

Current practice patterns of osteoporosis treatment in cancer patients and effects of therapeutic interventions in a tertiary center

PLOS ONE

Dear Dr. Fujihara,

Thank you for submitting your manuscript to PLOS ONE. After careful consideration, we feel that it has merit but does not fully meet PLOS ONE’s publication criteria as it currently stands. Therefore, we invite you to submit a revised version of the manuscript that addresses the points raised during the review process.

Reviewer 1 wishes a few additional details regarding drug treatment.  Reviewer 2 wishes a higher standard of English syntax and usage.  Editor agrees with both reviewers.

We look forward to receiving your revised manuscript.

Kind regards,

Robert Daniel Blank, MD, PhD

Academic Editor

PLOS ONE

Reviewers' comments:

Reviewer's Responses to Questions

**Comments to the Author**

1. If the authors have adequately addressed your comments raised in a previous round of review and you feel that this manuscript is now acceptable for publication, you may indicate that here to bypass the “Comments to the Author” section, enter your conflict of interest statement in the “Confidential to Editor” section, and submit your "Accept" recommendation.

Reviewer #1: (No Response)

Reviewer #2: (No Response)

2. Is the manuscript technically sound, and do the data support the conclusions?

Reviewer #1: Yes

Reviewer #2: Yes

3. Has the statistical analysis been performed appropriately and rigorously? 

Reviewer #1: Yes

Reviewer #2: Yes

4. Have the authors made all data underlying the findings in their manuscript fully available?

Reviewer #1: Yes

Reviewer #2: Yes

5. Is the manuscript presented in an intelligible fashion and written in standard English?

Reviewer #1: Yes

Reviewer #2: No

6. Review Comments to the Author

Reviewer #1: The result is somehow expected where those being treated should have a lower incidence of fragility fracture. It is supported by this piece of research. 9% of the patients were being treated with Vitamin D. Many current studies showed the important role of Vitamin D in patients with cancer especially breast cancer. Is Vitamin D considered a treatment for osteoporosis in patient with cancer in your centre? What kind of Vitamin D was being used? Was it single therapy or in combination with anti-osteoporosis medicines? Did you look at the vitamin D level of your patients, both baseline and after treatment?

Although it was mentioned in your document that there might be a role of teriparatide in treating cancer patients with osteoporosis, it is clearly stated in the teriparatide product information that it should not be used in cancer patients with skeletal metastatic lesion. What is your view on this?

Reviewer #2: The paper is scientifically sound and presents useful information. However, the written language used in the manuscript is not appropriate for a scientific publication (For example: first person used not be in scientific manuscript.

Editing of the paper by an individual scientist with English as their language would aid in acceptance of the paper.

7. PLOS authors have the option to publish the peer review history of their article (what does this mean?). If published, this will include your full peer review and any attached files.

Reviewer #1: No

Reviewer #2: No

---

## [Author Response · Author response to Decision Letter 1]

3 Feb 2021

February 3, 2021

Dear Dr. Blank,

Academic Editor 

PLOS ONE

Dear Editor: 

We would like to thank you for your response and for giving us the opportunity to improve and resubmit our manuscript (PONE-D-20-33343R1) entitled "Current practice patterns of osteoporosis treatment in cancer patients and effects of therapeutic interventions in a tertiary center." We are hereby resubmitting a revised manuscript conforming to all of the reviewers’ comments. In particular, we have addressed all the reviewers’ comments in a point-by-point manner and revisions are indicated in red font in the revised manuscript. We hope that the revised manuscript is now suitable for publication in your journal.

Thank you for your consideration. I look forward to hearing from you.

Sincerely,

Nasa Fujihara, 

Section of Orthopedic Surgery, Aichi Cancer Center 

1-1 Kanokoden Tikusa-ku, 

Nagoya City, Aichi, Japan 464-8681

Email: nfujihara@aichi-cc.jp

Phone: 052-762-6111

FAX: 052-762-6111

Reviewer #1: The result is somehow expected where those being treated should have a lower incidence of fragility fracture. It is supported by this piece of research. 9% of the patients were being treated with Vitamin D. Many current studies showed the important role of Vitamin D in patients with cancer especially breast cancer. Is Vitamin D considered a treatment for osteoporosis in patient with cancer in your centre? What kind of Vitamin D was being used? Was it single therapy or in combination with anti-osteoporosis medicines? Did you look at the vitamin D level of your patients, both baseline and after treatment?

Although it was mentioned in your document that there might be a role of teriparatide in treating cancer patients with osteoporosis, it is clearly stated in the teriparatide product information that it should not be used in cancer patients with skeletal metastatic lesion. What is your view on this?

Response

We would like to thank the reviewer for evaluating our manuscript and for these constructive comments. 

First, regarding vitamin D, we did use not the natural form of vitamin D that is often discussed nowadays. “Vitamin D” described in this study was actually “eldecalcitol” or “alfacalcidol,” which is an active vitamin D3 prodrug. Besides, we do not have a specific hospital or medical department protocol for applying Vitamin D drugs. Probably, the use of Vitamin D has been reported in some cases of breast cancer, as the reviewer stated. 

However, in Japan, “eldecalcitol” or “alfacalcidol” is often prescribed as an initial drug for osteoporosis treatment. Please note that we have edited the text and figures to make it clear that we used an active vitamin D3 prodrug. Especially, we have added the following sentences in the revised manuscript: 

“An active vitamin D3 form, the one most commonly prescribed in general practice, was used in 9% of all cases.” (Lines 137–138)

“The active form of vitamin D3 included eldecalcitol and alfacalcidol.” (Line 147)

Moreover, teriparatide's contraindication for cancer patients with skeletal metastatic lesion is mainly based on past studies showing that teriparatide increased osteosarcoma in rats. However, in this study on rats, the used doses were dozens of times higher and provided for a longer period of time than those given to humans. Therefore, there are many arguments regarding whether it can directly apply to humans. In fact, as a recent study that analyzed the data of patients from a US database did not show any evidence concerning the role of teriparatide in increasing osteosarcoma or malignant bone tumors, we believe that the indications for teriparatide will be expanded in the future. Alternatively, another new osteoporosis drug, romosozumab, is indicated for cancer patients and may be a useful treatment option. Please note that we have included these issues in the discussion of the revised manuscript as follows: “However, recent research has provided new evidence concerning the use of teriparatide (25, 30, 31). Interestingly, Gilsenan et al. reported that the incidence of osteosarcoma associated with teriparatide use during a 15-year period was not different than what was expected based on the background incidence rate of osteosarcoma after examining data from a US database (30). Besides, other new agents, such as romosozumab, humanized IgG2 monoclonal antibody, or annual zoledronic acid, were allowed to be used for the treatment of cancer patients. These relatively new drugs will be used more aggressively for such patients in the future.”

(Lines 256–263)

Reviewer #2: The paper is scientifically sound and presents useful information. However, the written language used in the manuscript is not appropriate for a scientific publication (For example: first person used not be in scientific manuscript.

Response

We would like to thank the reviewer for evaluating our manuscript and for this comment. Please note that we have sent our manuscript to an English editing company (Editage) for English proofreading. We hope that the level of English has significantly improved in the revised manuscript.

---

## [Decision Letter · Decision Letter 2]

22 Feb 2021

Current practice patterns of osteoporosis treatment in cancer patients and effects of therapeutic interventions in a tertiary center

PONE-D-20-33343R2

Dear Dr. Fujihara,

We’re pleased to inform you that your manuscript has been judged scientifically suitable for publication and will be formally accepted for publication once it meets all outstanding technical requirements.

Kind regards,

Robert Daniel Blank, MD, PhD

Academic Editor

PLOS ONE

Additional Editor Comments (optional):

Reviewers' comments:

Reviewer's Responses to Questions

**Comments to the Author**

1. If the authors have adequately addressed your comments raised in a previous round of review and you feel that this manuscript is now acceptable for publication, you may indicate that here to bypass the “Comments to the Author” section, enter your conflict of interest statement in the “Confidential to Editor” section, and submit your "Accept" recommendation.

Reviewer #1: All comments have been addressed

Reviewer #2: All comments have been addressed

2. Is the manuscript technically sound, and do the data support the conclusions?

Reviewer #1: Yes

Reviewer #2: Yes

3. Has the statistical analysis been performed appropriately and rigorously? 

Reviewer #1: Yes

Reviewer #2: Yes

4. Have the authors made all data underlying the findings in their manuscript fully available?

Reviewer #1: Yes

Reviewer #2: Yes

5. Is the manuscript presented in an intelligible fashion and written in standard English?

Reviewer #1: Yes

Reviewer #2: Yes

6. Review Comments to the Author

Reviewer #1: Change all DEXA to DXA. The question about the use of vitamin D has been well explained. Agreed with the input on the role of teriparatide.

Reviewer #2: (No Response)

7. PLOS authors have the option to publish the peer review history of their article (what does this mean?). If published, this will include your full peer review and any attached files.

Reviewer #1: No

Reviewer #2: No

---

## [Editor Report · Acceptance letter]

1 Mar 2021

PONE-D-20-33343R2 

Current practice patterns of osteoporosis treatment in cancer patients and effects of therapeutic interventions in a tertiary center 

Dear Dr. Fujihara:

I'm pleased to inform you that your manuscript has been deemed suitable for publication in PLOS ONE. Congratulations! Your manuscript is now with our production department. 

Kind regards, 

on behalf of

Professor Robert Daniel Blank 

Academic Editor

PLOS ONE